

# Decision support system for emergency management of oil spill accidents in the Mediterranean Sea

Svitlana Liubartseva[1], Giovanni Coppini[2], Nadia Pinardi[3], Michela De Dominicis[4], Rita Lecci[2], Giuseppe Turrisi[2], Sergio Creti[2], Sara Martinelli[2], Paola Agostini[2], Palmalisa Marra[5], and Francesco Palermo[2]

[1]Fondazione CMCC – Centro Euro-Mediterraneo sui Cambiamenti Climatici, via M. Franceschini 31, 40128 Bologna, Italy
[2]Fondazione CMCC – Centro Euro-Mediterraneo sui Cambiamenti Climatici, via Augusto Imperatore 16, 73100 Lecce, Italy
[3]Dipartimento di Fisica e Astronomia (DIFA) Universita degli Studi di Bologna, viale Berti-Pichat 6/2, 40127 Bologna, Italy
[4]National Oceanography Centre, 6 Brownlow Street, Liverpool, Merseyside L3 5DA, United Kingdom
[5]Links Management and Technology S.p.A. – via R. Scotellaro 55, 73100 Lecce, Italy

*Correspondence to:* S. Liubartseva (svitlana.liubartseva@cmcc.it)

**Abstract.** This paper presents an innovative web-based decision support system to facilitate emergency management in case of oil spill accidents, called WITOL (Where Is The Oil). The system can be applied to create a forecast of oil spill events, evaluate uncertainty of the predictions, and calculate hazards based on historical meteo-oceanographic datasets. To compute the oil transport and transformation WITOIL uses the MEDSLIK-II oil spill model forced by operational meteo-oceanographic

services. Results of the modeling are visualized through Google Maps. Special application for Android is designed to provide mobile access for competent authorities, technical and scientific institutions, and citizens.

## 1 Introduction

Possible oil spill accidents and operational pollution could have severe impacts on the Mediterranean basin. It is therefore crucial to provide the decision makers, stakeholders, and the public with trustworthy DSS (Decision Support System) able to

maintain the highest quality and near real-time information during oil pollution response.

New generation of DSS and similar systems are based on the environmental monitoring, state-of-the-art modeling, and innovative technology platforms. All of them incorporate the oil spill model as a forecasting tool driven by oceanographic and atmospheric datasets. Powerful data visualization module efficiently provides delivery service. Some DSS are connected with the environmental databases, evaluating the impact on population and ecosystem. Recently, several systems have been

set up and developed. Oil trajectory forecasting tool GNOME (Zelenke et al., 2012) and oil weathering model ADIOS (Lehr et al., 2002) were combined with the Environmental Sensitivity Index maps (Jensen et al., 1990) and a fine-resolution visualization module ERMA (ERMA, 2014) for efficient spill response planning (Muskat, 2014). At the European level, 4 oil spill models were integrated in the framework of MEDESS4MS Project (Mediterranean Decision Support System for Marine Safety) as follows: MOTHY (Daniel, 1996), MEDSLIK (Zodiatis et al., 2008), MEDSLIK-II (De Dominicis et al., 2013a),

and POSEIDON-OSM (Nittis et al., 2006). Using unified data flows, they were incorporated into a multi-model network to provide a round-the-clock forecast of oil spills. A similar combination of various global and regional meteo-oceanographic



forecasting systems with MOHID oil spill model (Carracedo et al., 2006) was conducted by Fernandes et al. (2013). Fusion of the SINTEF OSCAR oil spill model (Reed et al., 1995) with an environmental GIS dataset resulted in creation of DSS focused on estimation of economic and ecological impact (Wirtz and Liu, 2006). The system was applied to evaluate the short- and mid-term consequences of the $Prestige$ case (Spain, November 2002). Combination of SAR imagery, aerial surveillance and oil spill modeling by Seatrack Web (Ambjorn, 2007) was used to make appropriate decisions on oil combating activities in the Baltic and North seas (Anderson et al., 2010).

Up-to-date DSS requires not only the oil spill forecast but also evaluation of uncertainty of such a forecast, which is critical for timely, efficient and cost effective response and recovery. Uncertainty in prediction of the oil transport and transformation stems from the uncertain environment and data-sparse. In contrast to atmospheric pollution models (e.g., Bergin et al. (1999)), the methods of uncertainty quantifying are not well established in oil spill modeling. Nevertheless, uncertainty algorithms with respect to perturbation of ocean currents and wind were implemented in General NOAA Operational Modeling Environment (GNOME) (Zelenke et al., 2012). Currently, the majority of uncertainty estimations still remain a matter of research, being beyond the scope of web-based operational implementations. For example, Sebastiao and Guedes Soares (2007) varied the current and wind components, wind drag coefficient and deflection angle for the $Aragon$ case (Portugal, December 1989). Uncertainty analysis with respect to amount of spilled oil was carried out in Xu et al. (2012). Multi-model approach used an ensemble of meteo-oceanographic models was implemented in De Dominicis et al. (2014), which allowed the authors to improve significantly the quality of oil drift prediction.

In the present work, WITOIL (Where is The OIL) DSS (www.witoil.com) has been developed as a part of TESSA Project (Development of Technologies for the "Situational Sea Awareness") portfolio. The project aimed at strengthening the operational oceanography service in Southern Italy and integration with information platforms for delivering data on situational sea awareness. WITOIL is tailored to needs of managing the emergency situations and supporting the process of decision-making in response operations.

The distinct features of WITOIL include (1) 3-modular structure, embracing the oil spill forecast, uncertainty evaluation, and hazard assessment; (2) Lagrangian oil spill model MEDSLIK-II coupled with the basin-scale and regional operational meteo-oceanographic services; (3) web-based applications to different devices including personal computers, tablets, mobile phones, etc.; (4) visualization of geospatial information by means of Google Maps.

The manuscript is organized as follows: in Section 2, the oil spill model and meteo-oceanographic data are presented; Section 3 describes the structure and main components of DSS; and Sections 4 and 5 contain the results and conclusions, respectively.



## 2   Models and data

### 2.1   The oil spill model MEDSLIK-II

The oil spill model code MEDSLIK-II (De Dominicis et al., 2013a, b) is a freely available community model[1]. It is used to predict the transport and oil transformation due to complex physical processes occurring at the sea surface.

MEDSLIK-II calculates the advection-diffusion processes using a Lagrangian approach. The oil slick is discretized into constituent particles. Each particle moves due to currents, wind, and waves, which are provided by external basin-scale or regional oceanographic and atmospheric models.

The oil transformation processes at the surface are calculated by means of bulk formulas that describe the changes in the surface oil volume due to three main processes, known collectively as weathering (evaporation, dispersion and spreading).

Formation of water-in-oil emulsion is also taken into consideration (De Dominicis et al., 2013a). If an oil particle arrives on the coast, the model is able to simulate the adsorption of particles into the coastal environment taking into account a probability that oil may be washed back into the water.

As outputs, MEDSLIK-II provides the oil concentrations at the surface, in the dispersed water-column fraction, and on the coast. Mass balance components of the oil are calculated as a function of time, which allows temporal tracking the oil

weathering.

### 2.2   Meteo-oceanographic data

As detailed above, MEDSLIK-II requires the input of data about sea currents, sea surface temperatures and atmospheric wind.

The ocean current data and sea surface temperature are provided by hourly forecast and daily analysis fields produced by the Mediterranean Forecasting System (MFS) and by hourly forecast and daily simulation fields generated by the Adriatic

Forecasting System (AFS).

Covering the entire Mediterranean Sea with some extension into the Atlantic, MFS provides oceanographic data at 1/16°×1/16° horizontal resolution and 71 unevenly spaced vertical levels (Pinardi et al., 2003; Tonani et al., 2008). The analyses are produced by a data assimilation system that uses satellite and *in situ* data (Dobricic and Pinardi, 2008).

Adriatic forecasting system (Oddo et al., 2009) encompasses the whole Adriatic basin and extends southward of the Strait of

Otranto into the northern Ionian Sea (39.0–45.8° N 12.2–20.8° E). The system is implemented at the horizontal resolution of 1/45°×1/45° and 31 sigma layers along the vertical. The lateral boundary conditions for the sea current velocity, temperature, and salinity are imposed from MFS on a daily basis.

The atmospheric wind data are provided by the European Centre for Medium-Range Weather Forecasts (ECMWF) forecasts and analyses, at 0.125° horizontal and 6-hour temporal resolution.

---

[1] http://medslikii.bo.ingv.it/



## 3 Structure and main components of WITOIL DSS

Decision support system WITOIL consists of a client part and server component (Fig. 1). The client part is presented by GUI (Graphical User Interface) to configure and trigger off the system, as well to visualize the results using Google Maps.

The server component includes three modules as follows:

- oil spill forecasting module to provide prediction of the oil trajectory and fate on a near-real-time basis;

- uncertainty module aimed at on-the-fly estimation of uncertainty of the forecast; and

- hazard module for mapping probability of pollution from possible oil spills accidents on a long-term basis.

The server component incorporates MEDSLIK-II and meteo-oceanographic data sources described above. Hazard module uses pre-calculated MEDSLIK-II outputs based on the hypothetical oil spill scenarios and historical variability of the marine
environment simulated by the meteo-oceanographic models.

### 3.1 Oil spill forecasting module

Oil spill forecasting module provides resilient 24/7 operational service, supporting both basic and advanced use. In the basic mode, the system offers running the oil spill scenarios according to the initial information about the starting spill position and date, type of the oil, the oil volume or spill rate and duration, simulation length (the time period of tracking the oil spill) and
time step (output interval). Necessary oceanographic and atmospheric data can be chosen from a set of available model outputs.

Advanced use of WITOIL requires users who are experienced in oil spill modeling to set up advanced parameters and interpret results. A set of advanced parameters was selected from a full list of MEDSLIK-II parameters published in De Dominicis et al. (2013a). Parameters that can be varied by advanced users are listed in Tab. 1. Stokes drift and a wind drift factor were rated as being of much importance, and were highly discussed in literature (Lehr et al., 2002; De Dominicis et al.,
2013a). Horizontal diffusivity is responsible for spatial extension of the oil spill in time. Three parameters that mainly control the weathering processes (evaporation, dispersion and spreading) can be also modified according to the user's needs. Number of oil particles represented the oil slick can be also changed to control the Lagrangian discretization of the oil slick. Finer discretization results in the smoother oil spill patterns, but it requires more computational time.

### 3.2 Uncertainty module

In addition to changing the advanced parameters manually and tracking the differences, WITOIL provides automatically calculation of uncertainties with respect to some initial conditions of the oil spill. Currently, there is not a common approach to calculations of uncertainties in oil spill modeling. Our methodology is based on of a parametric analysis, typical of the atmospheric Lagrangian models (Bergin et al., 1999). Due to a large number of parameters that control the oil movement and transformation in MEDSLIK-II, the number of possible uncertainty scenarios is enormous. WITOIL allows users to evaluate
online the uncertainty arisen from unknown initial conditions: (1) starting position, (2) starting date, (3) starting and final dates,




that can be varied within user-specified ranges (Tab. 2). The spread obtained in oil concentrations and represented in probability terms can be interpreted as a measure of uncertainty.

In $Scenario\ I$, uncertainty caused by unknown initial oil spill position is calculated in case of so called "cross" and "hexagon" configurations (Fig. 2). For any location of interest $(x^0, y^0)$ inside the uncertainty radius $R$, the set of spills with the

spill indices of $i = 1, 2, ..., N$ is generated. In case of "cross" configuration $N = 5$. In case of "hexagon" configuration $N = 7$. All the spills begin at the same time, and they finish synchronously. The final oil concentrations $C^i(x, y)$ are aggregated to produce a dimensionless concentration functions $H^i(x, y)$ defined as:

$$H^i(x, y) = \begin{cases} 1, & C^i(x, y) > C_{thr} \\ 0, & C^i(x, y) \leq C_{thr} \end{cases}, \tag{1}$$

where $C_{thr}$ is the threshold oil concentration, specified by users. For the examples described below, we assume that $C_{thr} = 0$.

A function $P(x, y)$ expressed in probability terms (%) can be used as a measure of uncertainty:

$$P(x, y) = 100\% \frac{1}{N} \sum_{i=1}^{N} H^i(x, y). \tag{2}$$

Values of $P(x, y)$ range between 0–100%, indicating the area where exceeding the threshold concentration is the most probable.

Although the "hexagon" configuration produces slightly smoother probability patterns than the "cross" ones, the latter is quicker to calculate.

In $Scenario\ II$, uncertainty caused by unknown starting date of the oil spill is calculated. Hence, for the point of $(x^0, y^0)$, a set of oil spills with the spill indices of $i = 1, 2, ..., N$ are generated inside the uncertainty interval $T^0$ with respect to the starting date of $t^0$. As Fig. 3 indicates, initial dates of these spills are uniformly shifted relative to each other. At the time of interest $t^*$, the final oil concentrations $C^i(x, y)$ are aggregated to produce a dimensionless concentration functions $H^i(x, y)$ due to Eq. (1). In Fig. 3, the grey boxes depict the final oil concentrations. As the result, the probability function $P(x, y)$ calculated by means

of Eq. (2) can be interpreted as a measure of the uncertainty regarding the initial date of the oil spill.

In $Scenario\ III$, a set of spills is also initiated for the location of $(x^0, y^0)$. Apart from variation of the starting date $t^0$ within the range of $T^0$, the final date of the spillage $t^*$ varies within the uncertainty interval $T^*$ (Fig. 4). As Fig. 4 shows, both the initial and final dates are shifted relative to each other (in contrast to $Scenario\ II$, where the simulation length is frozen for all the spills). To calculate $P(x, y)$ Eqs. (1), (2) are applied.

**3.3   Hazard module**

Hazard module is tailored to plan response strategy and increase the level of preparedness prior to the real oil spill accidents. Hazard maps represent statistically the MEDSLIK-II outputs basing on historical variability of the meteo-oceanographic data over a long-term period (Liubartseva et al., 2015). This information can be also used for optimization in deployment of spill response equipment and training the response personnel.

Nowadays, accidents on offshore oil and gas facilities are mentioned among risks that might result in severe marine pollution in the Mediterranean Sea. In present work, we calculated hazard maps in case of potential accidents on six Adriatic oil plat-



forms: *Carmen*, *Dora*, *Elisa*, *Elsa*, *Gilga*, and *Ombrina* (Fig. 5). To perform the calculations oil spill model MEDSLIK-II was coupled with the daily AFS simulations and 6-hour ECMWF analyses over the 7 year time period (2007–2013). On a daily basis, we initiated a typical oil platform leak scenario represented a failure in the oil transportation system (Zhuravel et al., 2013):

- continuous oil spill with duration of 120 h;

- spill rate of 80 ton hour$^{-1}$;

- simulation length of 240 h; and

- oil density of 808 kg m$^{-3}$ (API=17).

Focusing on accidental oil platform leaks, we used the relative high threshold concentrations as follows, 1 ton km$^{-2}$ at the sea
surface; 0.1 ton km$^{-1}$ on the coastline; and 1 ton km$^{-2}$ for the dispersed oil fraction (French McCay et al., 2004).

## 4 Results

Service-oriented approach plays a key role in the WITOIL DSS development. The 8-language graphical user interface is supplied with a great variety of user services including help support, tooltips, a video tutorial[2], *etc.* The system is designed to meet the real-time requirements in terms of performance and dynamic service delivery. Comprehensive computational resources
incorporated in WITOIL DSS and network bandwidth efficiently support multi-user regime and independently processing the user requests.

### 4.1 Oil spill forecast

An example of using the oil spill forecasting module is shown in Fig. 6. Information about hypothetical oil spill located in the Southern Adriatic Sea (41.36° N 17.50° E) near the shipping lane Ancona (Italy) – Igoumenitsa (Greece), was put into
20 WITOIL DSS. The accident started on 16 February 2015 at midnight. It was a continuous spill with a rate of 80 ton hour$^{-1}$ and expected duration of 72 h. The oil type of Aboozar with API of 26.96 and density of 893 kg m$^{-3}$ was spilled. Oil spill forecast was carried out by means of the ECMWF and MFS meteo-oceanographic datasets. The corresponding field of sea surface currents, underlying an area of the accident, indicated that initially, the spill was located in the region of relatively weak currents at the periphery of the South Adriatic Gyre (Artegiani et al., 1997). The sea surface current field presented in
Fig. 6 is a product of MFS visualized by means of SEACONDITIONS[3], one of the services in the TESSA Project portfolio.

Following the user's settings, WITOIL produced the hourly snapshots of oil concentration at the surface and on the coastline, and their animation. Recent results calculations can be automatically saved at the server side and quickly restored afterwards. In Fig. 7, the last snapshot of the oil spill forecast is presented. In 72 hours after the accident, an elongated oil spill moved

---

[2]https://www.youtube.com/watch?v=qj_GokYy8MU
[3]http://www.sea-conditions.com





under the influence of the local currents and Stokes drift. Wind speed vector with a magnitude of about $4\,\mathrm{m\,s^{-1}}$ m in the slick mass center is depicted in Fig. 7. Local currents were rather weak and moderate, and they demonstrated cyclonic vorticity in the slick area. It is clearly evident that the South Adriatic Gyre directly influenced the transport and shape of the spill. Oil concentration at the sea surface reached $70\,\mathrm{ton\,km^{-2}}$, while the coastline of Italy remained pristine.

## 4.2   Uncertainty estimations

After receiving the forecast of the oil spill movement, uncertainty of this forecast can be obtained online. First of all, uncertainty with respect to starting oil spill position was calculated in the framework of $Scenario\ I$ (Fig. 2) described above. Uncertainty radius of $R = 2$ miles was chosen as an example. As Fig. 8 indicates, south and southeast directions at the distance of about 8 miles were found to be the most probable (> 75%) for the oil location in 72 h after the accident. Besides, there was a small but non-zero probability (about 20%) that oil went west of the starting position. Both "cross" and "hexagon" configurations gave very similar probability distributions. In case of the "cross" configuration (Fig. 8a), the probabilities indicated 5 gradations: 20 $(1/5 \cdot 100\%)$, 40 $(2/5 \cdot 100\%)$, 60 $(3/5 \cdot 100\%)$, 80 $(4/5 \cdot 100\%)$, and 100 $(5/5 \cdot 100\%)$. In case of the "hexagon" configuration (Fig. 8b), they varied among 7 gradations: $1/7 \cdot 100\%, 2/7 \cdot 100\%, 3/7 \cdot 100\%, 4/7 \cdot 100\%, 5/7 \cdot 100\%, 6/7 \cdot 100\%$, and $7/7 \cdot 100\%$.

An example of the uncertainty evaluation with respect to starting date of oil spill is presented in Fig. 9. In $Scenario\ II$ (Fig. 3), 7 spills were run inside the uncertainty interval of $T^0 = 36$ h. The spills were shifted by 6 h relative to each other. Probability distribution indicated a narrow comet-shaped band with the highest values at the distance of 3–8 miles southwest and southeast of the starting position. In Fig. 9, the probability spread was not significant, which allowed coming to the conclusion that, for variations of starting spill time in the interval of 36 h, uncertainty in the oil spill forecast was not high.

Seven spills were also initiated inside the uncertainty intervals of $T^0 = T^* = 36$ h in $Scenario\ III$ (Fig. 4). The spills were shifted by 6 h. In contrast to $Scenario\ II$, the obtained probability distribution demonstrated more significant spatial spread. As in Fig. 10 shown, the most probable (> 71%) direction was southwest, while the southeast one was less probable (about 14%). It can be concluded that, in the aforementioned cases, the main forecast uncertainty was arisen from the degree of the South Adriatic Gyre influence.

## 4.3   Hazards from the oil platforms

To calculate hazards from potential accidents on six Adriatic oil platforms ($Gilda$, $Carmen$, $Elisa$, $Dora$, $Elsa$ and $Ombrina$) over 15,000 MEDSLIK-II runs were carried out for the 2007–2013 time period. In general, more than 1500 monthly hazard maps were produced including the hazard maps at sea surface, on the coastline and the maps for the dispersed oil fraction. Additionally, more than 400 averaged hazard maps were calculated: yearly, seasonally averaged maps and monthly climatology maps. Currently, hazard module has not yet incorporated in WITOIL DSS. However, the further version of WITOIL will be able to sort out and visualize all the pre-calculated hazard maps.

Two oil platforms, $Carmen$ and $Ombrina$, were found to be the most hazardous to the Italian coastal zone. Monthly averaged maps Fig. 11 demonstrated the most significant spatial variability of hazards.



As Fig. 11a, d indicate, the shape of the highest surface hazard area followed the climatological shape of the Western Adriatic Coastal Current (Artegiani et al., 1997; Oddo et al., 2005). The same feature was found on the hazard maps for dispersed fraction of the oil, but at lower level of hazards (Fig. 11c, f). For both platforms, the maximum magnitude of hazard on the coastline tended to exceed the values at the sea surface and in the water column. It means that in case of accidents on the oil platforms the most hazardous impact is expected on the Italian coastline.

5  In case of $Carmen$, hazards reached the maximum values of more than 0.8 on the coastline from Porto Sant'Elpidio to Porto San Giorgio (Fig. 11b). Sea surface hazard distributions revealed high hazards (> 0.5) near Porto Sant'Elpidio, while a narrow area of relatively moderate hazards (> 0.3) extended up to Pescara (Fig. 11a). Hazard from dispersed oil fraction were rather low (0.05–0.25) but due to a long-term cumulative effect, some impact on fishery stocks near Pescara and Ortona (Fig. 11c) might be expected.

10  Possible accidents on $Ombrina$ lead to high coastline hazards (> 0.8) in Zona Industriale Porto di Vasto and Termoli, covering the north coast of the Gargano Peninsula (Fig. 11e), where Gargano National Park is situated. As well coastline of the Island Tremiti (14 miles north off the Gargano Peninsula, 42.12° N 15.51° E) also indicated significant hazards (0.5–0.7). Sea surface hazards showed high values (> 0.5) in the area between the platform and Termoli (Fig. 11d). Hazards from the dispersed oil were rather low (0.05–0.25) but they was be able to impact the diving sites near the Tremiti Archipelago (Fig. 11f).

## 4.4 Mobile application for Android

A simplified version of WITOIL DSS is developed for Android to share information during field response operations (Fig. 12). User-friendly interface incorporates the necessary set of general parameters in order to obtain the oil spill forecast, make decisions and disseminate them among the members of response team on a near-real-time basis. In addition, users may take advantage of unified access to all the mobile applications of TESSA services. For example, SEACONDITIONS provide access 20 to forecast of surface currents, sea surface temperature, significant wave height and direction, wave period and direction; air temperature, surface pressure, precipitation, cloud coverage, wind speed, etc. WITIOL application for iOS is planned to develop in the next version of the system.

## 5 Conclusions

WITOIL decision support system as the innovative web-based tool has been implemented in the framework of TESSA project. 25 The system is designed to support emergency organization in case of oil spill accidents in the Mediterranean Sea.

 The system incorporates three modules, namely oil spill forecast, uncertainty evaluation, and hazard assessment. WITOIL is based on MEDSLIK-II oil spill code (De Dominicis et al., 2013a, b) coupled with the operational oceanographic and atmospheric services. The system allows representation of the model predictions and geospatial data on different types of devices: personal computers, tablets, mobile phones, $etc.$

30 Oil spill forecast module focuses on performing reliable and accurate predictions of oil spill trajectory and fate, which can be routinely updated basing on operational forecasting chain.





Uncertainty module allows users to evaluate variability of the forecast caused by uncertain data on the initial oil spill conditions.

Hazard module identifies the impact areas (at the sea surface, on the coastline and in the water column) in case of possible oil spill accidents, basing on the Lagrangian modeling the oil drift driven by historical meteo-oceanographic datasets. Calculation of hazards requires significant computational resources. WITOIL will be used for visualization of pre-calculated hazard maps.

Functionality of the system has been demonstrated by the example of hypothetical oil spill in the Southern Adriatic Sea in February 2015. The 72-hour forecast indicated initial southwestward movement of the oil with subsequent southeastward transport by the South Adriatic Gyre. However, uncertainty analysis with respect to initial location and date of the oil spill showed that the oil spill trajectory could shift eastward or westward depending on the influence of the South Adriatic Gyre.

Capability of the hazard module has been illustrated by hazard maps for six Adriatic oil platforms in case of potential accidents due to failure in the oil transportation system. $Carmen$ and $Ombrina$ oil platforms were found to be the most hazardous to the Italian coastal zone. Distinctive "hot spots" were indicated along the coast from Porto Sant'Elpidio to Porto San Giorgio; in Zona Industriale Porto di Vasto and Termoli, as well on the north coastline of the Gargano Peninsula. WITOIL application to Android has been implemented to share oil spill forecast during the field response activities.

In current version of WITOIL DDS, oil forecasting module is the most advanced one due to the highest priority. Its development is based on all the previous experience of MEDSLIK-II applications to the real cases (De Dominicis et al., 2013b, 2014; Coppini et al., 2011; Alves et al., 2015). Further implementation of the forecasting module will be related to the improvement of integration with the other high-resolution meteo-oceanographic models and user services. To our knowledge, the developed uncertainty module regarding the initial oil spill conditions is unique. Therefore, users' feedback is needed for the further implementation of this module. Future progress in hazard module can be associated with gaining the MEDSLIK-II oil spill archive to involve additional oil spill sources and cover the whole Mediterranean Basin.

*Acknowledgements.* This work was performed in the framework of the TESSA Project (Sviluppo di TEcnologie per la Situational Sea Awareness) supported by PON (Ricerca & Competitivita 2007–2013) cofunded by UE (Fondo Europeo di sviluppo regionale), MIUR (Ministero Italiano dell'Universita e della Ricerca), and MSE (Ministero dello Sviluppo Economico). The authors thank Angela Cocozza for performing hazard calculations for $Ombrina$ oil platform.



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



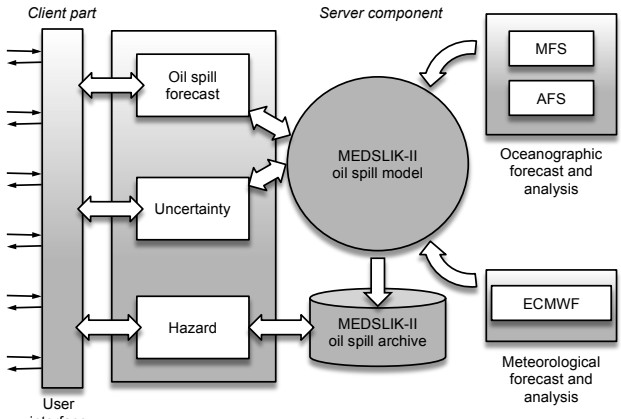

**Figure 1.** Schematic structure of WITOIL DSS

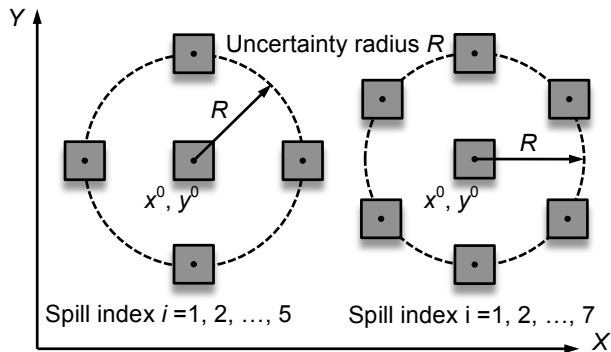

**Figure 2.** Uncertainty $Scenario\ I$ with respect to initial position of the oil spill: "cross" (left panel), and "hexagon" configuration (right panel)

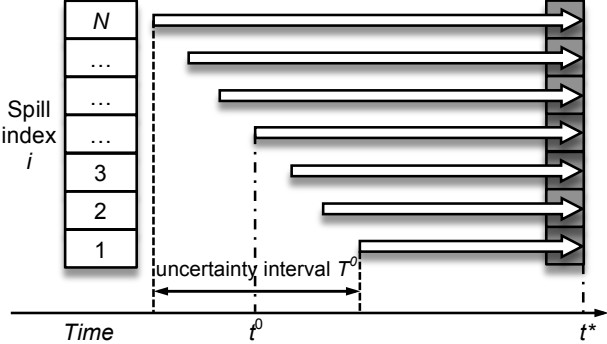

**Figure 3.** Uncertainty $Scenario\ II$ with respect to the initial date of the oil spill





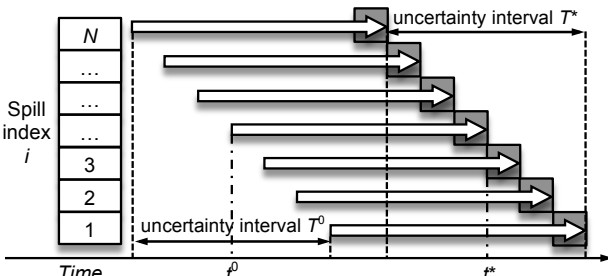

**Figure 4.** Uncertainty *Scenario III* with respect to the initial and final dates of the oil spill

**Table 1.** Parameters for advanced user specification

| ## | Parameter | Default value | Physical meaning | Discussed in reference |
|---|---|---|---|---|
| #1 | Stokes drift | Yes | Turns on the Stokes drift | (De Dominicis et al., 2013a) |
| #2 | Direct windage effect | No | Turns off the windage effect | (Lehr et al., 2002) |
| #3 | Horizontal diffusivity | $2.0 \, \mathrm{m^2 \, s^{-1}}$ | Controls horizontal turbulent dispersion | (De Dominicis et al., 2013a) |
| #4 | Specific evaporation rate | $3.3 \cdot 10^{-5} \, \mathrm{m \, s^{-1}}$ | Controls evaporation of the oil | (Mackay et al., 1979) |
| #5 | Specific dispersion rate | $8.0 \cdot 10^{-6} \, \mathrm{s^{-1}}$ | Controls dispersion of the oil by breaking waves | (Mackay et al., 1979) |
| #6 | Specific rate of spreading of thick slick | $150 \, \mathrm{s^{-1}}$ | Controls gravity-viscous spreading of the oil | (Al-Rabeh et al., 2000) |
| #7 | Number of oil particles | 90 000 | Represents Lagrangian discretization of the slick | (De Dominicis et al., 2013b) |

**Table 2.** Uncertainty scenarios implemented in WITOIL

| | Type of uncertainty | Variable parameters | Range of variation | Reference figure |
|---|---|---|---|---|
| *Scenario I* | Spatial | Initial position of the oil spill | Uncertainty radius $R$ | Fig. 2 |
| *Scenario II* | Temporal | Initial date of the oil spill | Uncertainty interval with respect to initial date of the spill $t^0$ | Fig. 3 |
| *Scenario III* | Temporal | Initial and final dates of the oil spill | Uncertainty interval with respect to initial date of the spill $t^0$ and final date $t^*$ | Fig. 4 |



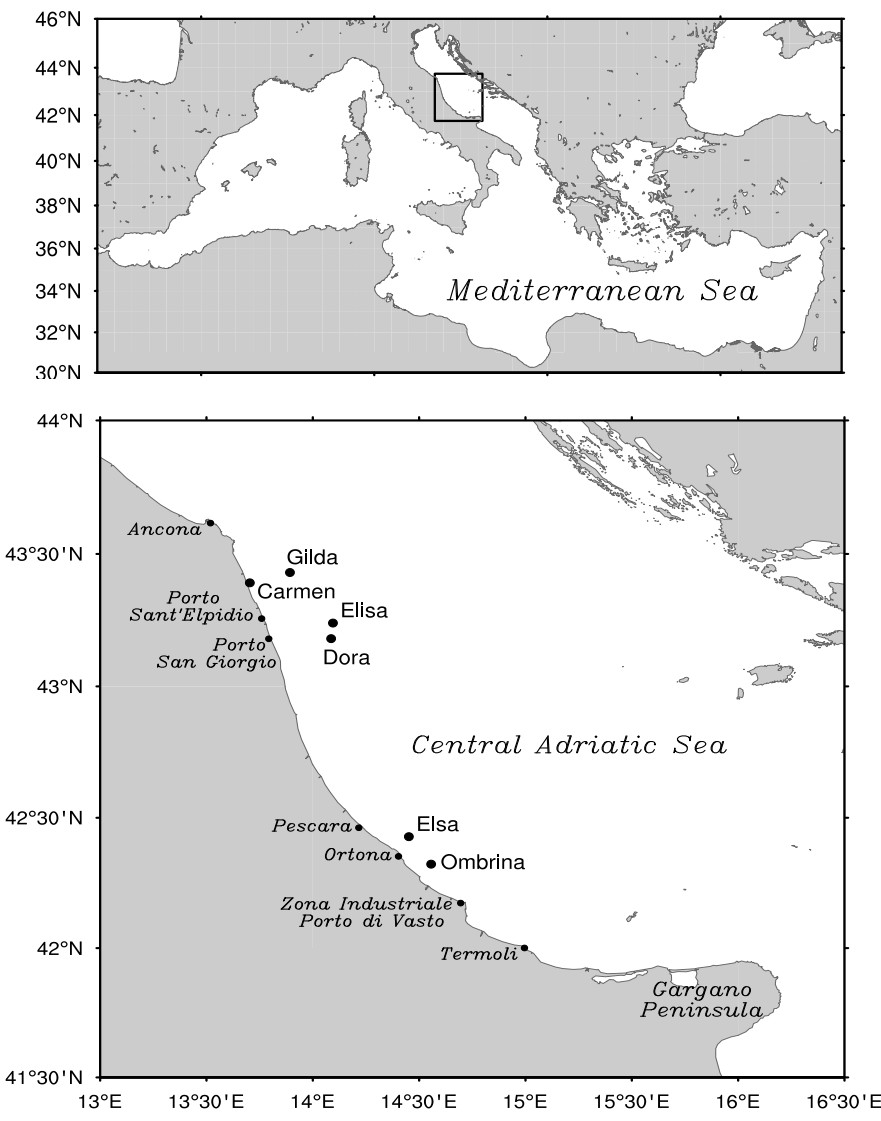

**Figure 5.** The Adriatic sub-domain, where hazards from potential accidents on the oil platforms (*Gilda*, *Carmen*, *Elisa*, *Dora*, *Elsa* and *Ombrina*) were calculated




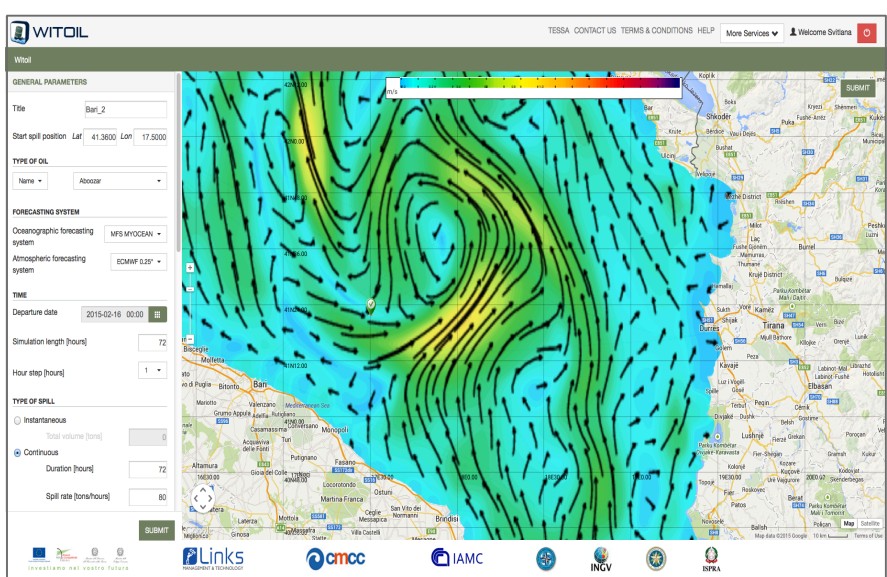

**Figure 6.** User interface of WITOIL decision support system

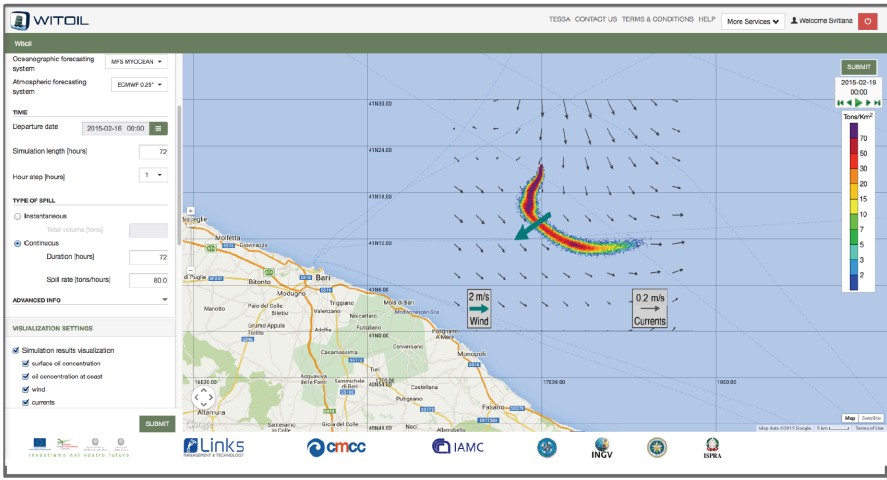

**Figure 7.** Result of the oil spill forecast in 72 hours after the accident





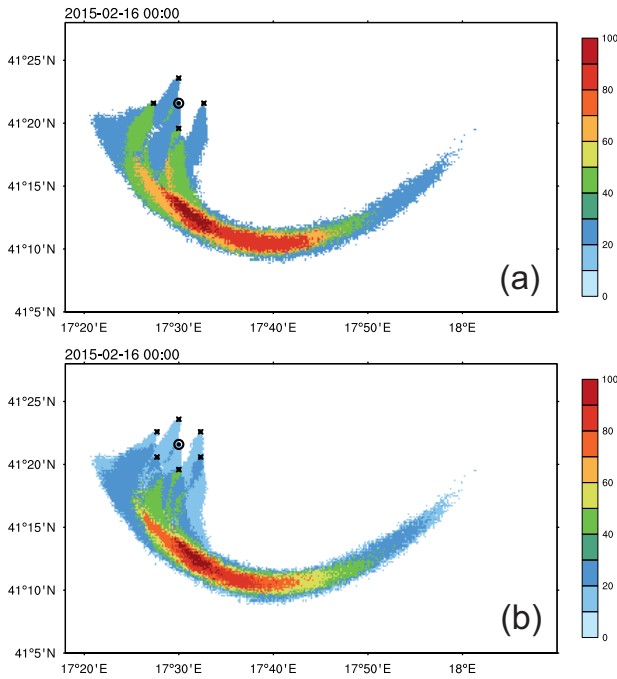

**Figure 8.** Uncertainty of the oil spill forecast with respect to starting location of the oil spill: in "cross" (a) and "hexagon" (b) configuration

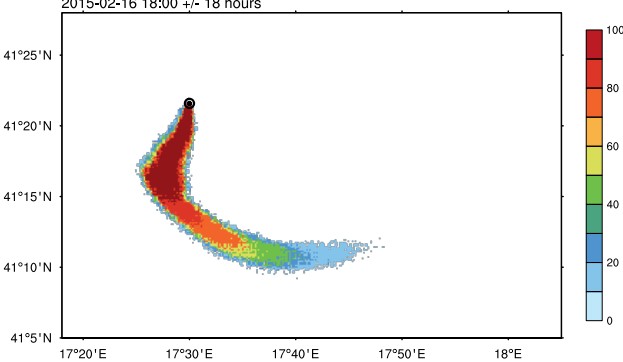

**Figure 9.** Uncertainty of the oil spill forecast with respect to starting date of the oil spill





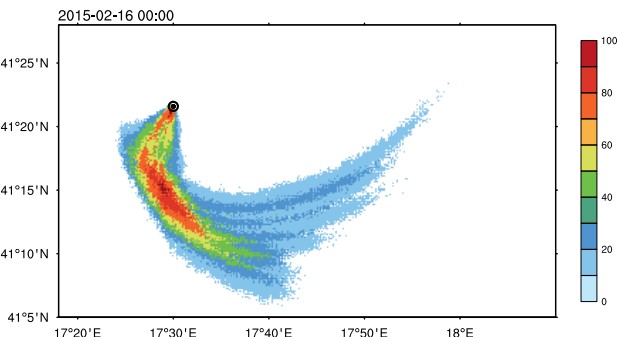

**Figure 10.** Uncertainty of the oil spill forecast with respect to starting and final dates of the oil spill




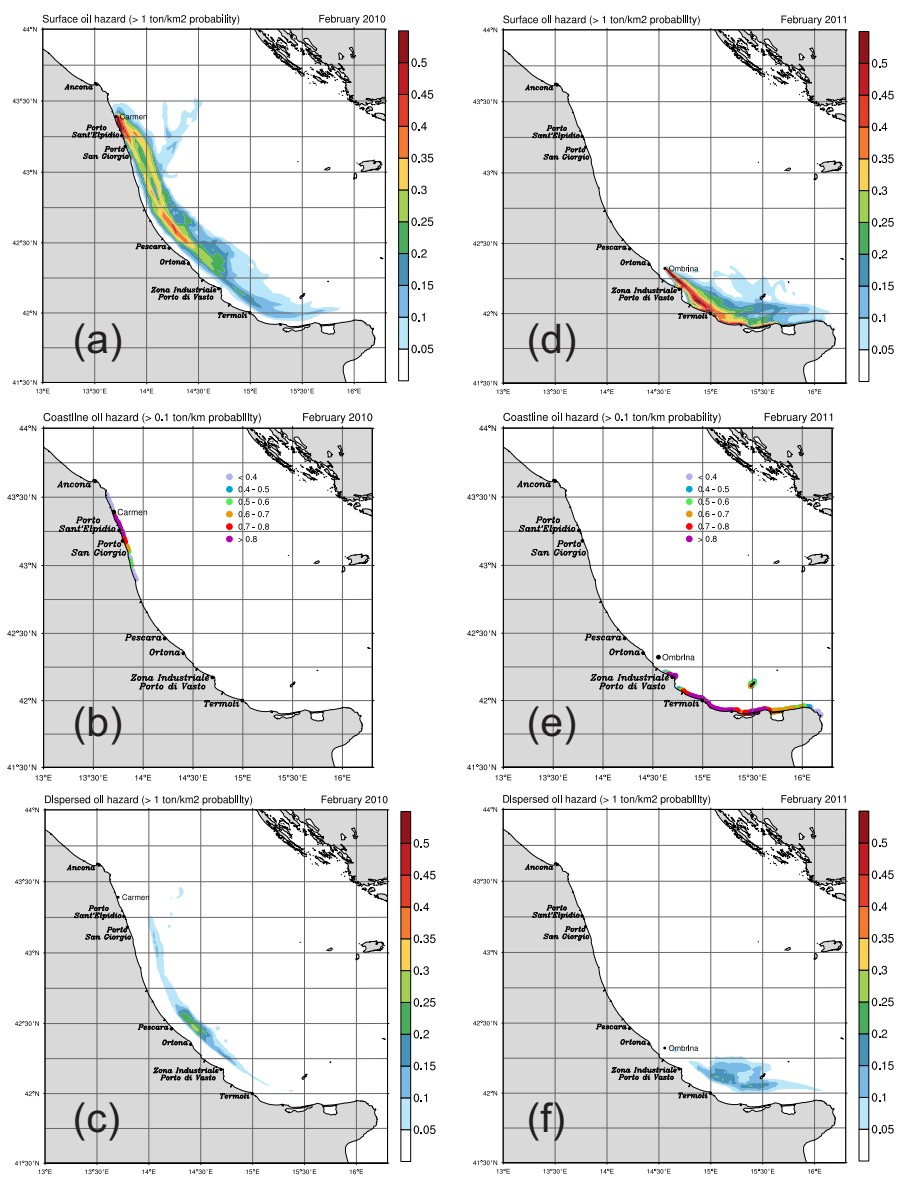

**Figure 11.** Monthly averaged hazard maps for the oil platform of $Carmen$ in February 2010 (left panel): at the sea surface (a), on the coastline (b), for the dispersed oil (c); and $Ombrina$ oil platform in February 2011 (right panel): at the sea surface (d), on the coastline (e), for the dispersed oil (f)




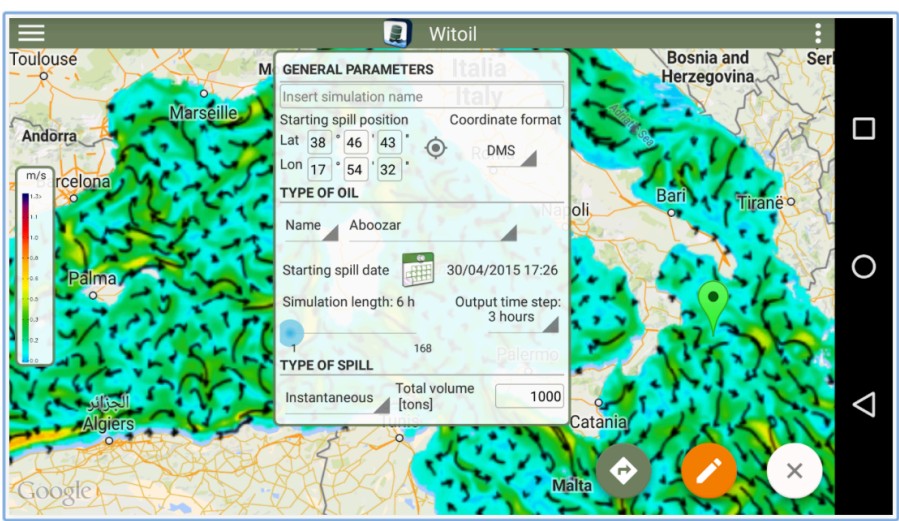

**Figure 12.** General view of WITOIL application for Android