# Peer review of "Decision support system for emergency management of oil spill accidents in the Mediterranean Sea"

_Natural Hazards and Earth System Sciences, 2016_

## Referee Comment (RC1) · Anonymous Referee #1 · 30 Jun 2016

The paper deals with web based DDS developed under the framwork of the TESSA project, in support of the emergency management of the oil spills in the Mediterranean Sea. The paper is well organized and well written. A review of the Medslick II is presented and description of the WITOIL DSS presented with one application to an hypotetycal spill in the Southern Adriatic.

I suggest to following reference to the MedslickII code in 2.1 and in the conclusions at line 16 page 9:

Samaras, Achilleas G.; De Dominicis, Michela; Archetti, Renata; Lamberti, Alberto; Pinardi, Nadia. Towards improving the representation of beaching in oil spill models: A case study. MARINE POLLUTION BULLETIN Volume: 88 Issue: 1-2 Pages: 91-101 DOI: 10.1016/j.marpolbul.2014.09.019. 2014.

---

## Author Comment (AC1) · 30 Jun 2016

Dear Anonymous Referee, thank you very much for your comments. I really appreciate them; particularly, your idea to add the reference to a very good paper by Samaras et. all (2014). Not only does this publication include the simulations of a real Lebanon case but also brings about the progress in the beaching formalism in MEDSLIK-II. I have already added the reference to the next, corrected version of the manuscript. Best regards, on behalf of the co-authors, Svitlana Liubartseva

---

## Referee Comment (RC2) · Anonymous Referee #2 · 6 Jul 2016

This paper presents a web-based decision support system for emergency management in case of oil spill accidents in the Mediterranean Sea, called WITOIL (Where Is The Oil). Such systems already exist in other areas, for example Seatrack Web in the Baltic Sea, or were developed in projects such as MEDESS-4MS.

The system is based on the MEDSLIK-II oil spill model and includes an uncertainty algorithm for the initial conditions. It includes also a hazard module based on historical meteo-oceanographic datasets with an application to six Adriatic oil platforms.

The system is clearly design for a service-oriented approach with a friendly Graphical User Interface with results visualized through Google Maps. It has also a mobile version for Android.

I tried to run a simulation from the web interface (www.witoil.com). By default, the

velocity due to wave-induced currents or Stokes drift is set to on, but there is no wave model to choose. It probably means that MEDSLIK-II uses an analytical formulation that depends on wind amplitude rather than wave data from a wave model, but that information is not clearly stated.

The paper is well organized and well written. Figures are very informative and clear. It is my opinion that this work is appropriate for NHESS.

Typo error to be checked: page 1, line2: replace "WITOL" by "WITOIL"

---

## Author Comment (AC2) · 7 Jul 2016

Thank you very much for your very relevant comment on the Stokes drift calculation. To the final variant of the manuscript, we have added the paragraph below:

To calculate the Stokes drift, the latest version of WITOIL uses empirical, so called JONSWAP wave spectrum as a function of wind speed and fetch (Hasselmann et al., 1973). Currently, De Dominicis et al., (2016) have modified MEDSLIK-II for the direct usage of wave model outputs, which is more accurate and computationally efficient. This important capability will be adapted into the next version of WITOIL.

and two additional references:

Hasselmann, K., Barnett, T., Bouws, E., Carlson, H., Cartwright, D., Enke, K., Ewing,

[Figure]

J., Gienapp, H., Hasselmann, D., Kruseman, P., Meerburg, A., Mller, P., Olbers, D., Richter, K., Sell, W., Walden, H., 1973. Measurements of wave growth and swell decay during the Joint North Sea Wave Project (JONSWAP). Ergnzungsheft zur Deutschen Hydrographischen Zeitschrift Reihe, A8–12.

De Dominicis, M., Bruciaferri, D., Gerin, R., Pinardi, N., Poulain, P.M., Garreau, .P., Zodiatis, G., Perivoliotis, L., Fazioli, L., Sorgente, R., Manganiello, C., 2016. A multi-model assessment of the impact of currents, waves and wind in modelling surface drifters and oil spill. Deep-Sea Res. II, http://dx.doi.org/10.1016/j.dsr2.2016.04.002i

We also appreciate your finding the typo. Now, we have corrected it.
* * *